# Effectiveness of Dimple Microtextured Copper Substrate on Performance of Sn-0.7Cu Solder Alloy

**DOI:** 10.3390/ma16010096

**Published:** 2022-12-22

**Authors:** Siti Faqihah Roduan, Juyana A. Wahab, Mohd Arif Anuar Mohd Salleh, Nurul Aida Husna Mohd Mahayuddin, Mohd Mustafa Al Bakri Abdullah, Aiman Bin Mohd Halil, Amira Qistina Syamimi Zaifuddin, Mahadzir Ishak Muhammad, Andrei Victor Sandu, Mădălina Simona Baltatu, Petrica Vizureanu

**Affiliations:** 1Faculty of Chemical Engineering and Technology, Kompleks Pusat Pengajian Jejawi 2, Universiti Malaysia Perlis (UniMAP), Arau 02600, Malaysia; 2Geopolymer and Green Technology, Centre of Excellence (CEGeoGTech), Universiti Malaysia Perlis (UniMAP), Arau 02600, Malaysia; 3Faculty of Mechanical and Automotive Engineering Technology, Universiti Malaysia Pahang, Pekan 26600, Malaysia; 4Department of Technologies and Equipments for Materials Processing, Faculty of Materials Science and Engineering, Gheorghe Asachi Technical University of Iaşi, Blvd. Mangeron, No. 51, 700050 Iasi, Romania; 5Romanian Inventors Forum, Str. Sf. P. Movila 3, 700089 Iasi, Romania; 6National Institute for Research and Development in Environmental Protection INCDPM, Splaiul Independentei 294, 060031 Bucharest, Romania; 7Technical Sciences Academy of Romania, Dacia Blvd. 26, 030167 Bucharest, Romania

**Keywords:** laser surface texturing, dimple microtexture, lead-free solder, wettability, intermetallic compound

## Abstract

This paper elucidates the influence of dimple-microtextured copper substrate on the performance of Sn-0.7Cu solder alloy. A dimple with a diameter of 50 µm was produced by varying the dimple depth using different laser scanning repetitions, while the dimple spacing was fixed for each sample at 100 µm. The dimple-microtextured copper substrate was joined with Sn-0.7Cu solder alloy using the reflow soldering process. The solder joints’ wettability, microstructure, and growth of its intermetallic compound (IMC) layer were analysed to determine the influence of the dimple-microtextured copper substrate on the performance of the Sn-0.7Cu solder alloy. It was observed that increasing laser scan repetitions increased the dimples’ depth, resulting in higher surface roughness. In terms of soldering performance, it was seen that the solder joints’ average contact angle decreased with increasing dimple depth, while the average IMC thickness increased as the dimple depth increased. The copper element was more evenly distributed for the dimple-micro-textured copper substrate than its non-textured counterpart.

## 1. Introduction

Biomimetics, also known as biomimicry, is the imitation of nature’s movements, materials, and processes. Nature observance and the knowledge gained can be utilised to form new ideas. Engineers have utilised biomimicry to develop and improve new products by imitating the characteristics of plants and animals, and biomimicry products are now everywhere, including everyday items.

An example of a nature-inspired product is Velcro, which owed its genesis to the behaviour of cockleburs sticking to animals and clothes. Cockleburs possess tiny hooks that allow them to adhere tightly to the loops of fur or fabric. Other well-known products are window glass, self-cleaning exterior paint, and umbrella fabric. These products were mimicked from the lotus flower plant, which is made up of tiny bumps covered by waxy crystals to produce a self-cleaning mechanism [1]. Most natural organisms possess characteristic micro/nano-texture designed to ensure their survival and adaptation. These microtextures have been researched extensively in various applications and, when possible, exploited to enhance materials’ performance, specifically tribology, friction, and wear [2,3,4].

Textures are usually tailored per their intended effects. Some typical surface textures include grids, riblets, grooves, and dimples. An example is a golf ball with dimples on its surface. The dimples are designed to enhance its aerodynamic performance [5] by producing a thin, turbulent boundary layer of air that clings to the surface. This permits the air to flow around the ball’s surface, culminating at the rear side of the ball, which minimises the air resistance that opposes the golf ball’s flight direction.

A study in brazing application has found that the convex platform and groove-textured surfaces fabricated on the Ti_3_SiC_2_ ceramic surface effectively decreased the wetting angle of the brazing filler metal from 59.6° to 25.7° [6]. According to the research, the brazing filler metal fills the microstructural groove via a capillary action mechanism. Another reported observation was that the Ti3SiC2 edge’s diffusion layer width was more significant when it had grooves on its surface. This was because grooves facilitate the migration of Ag and Cu atoms into the Ti3SiC2 ceramic and directly improve the diffusion mechanism during the brazing process.

Li et al. [7] manufactured micro-grooves on Ti_6_Al_4_V substrates and reported that the wettability was enhanced only when the groove spacing exceeded 0.25 mm. Textured surfaces are commonly used to improve the adhesion between solid substrate surfaces.

Zhiyang Liu et al. [8] fabricated micro/nano-ripples, micro-grooves, and micro-pits on stainless steel using an ultrafast laser to study the wetting and spreading behaviours of aluminium-silicon (Al-Si). They reported that the fabricated micro/nano-ripples successfully decreased the contact angle from 10.8° to 8.7°, enhancing the Al-Si’s wettability. They also reported that the thickness of the reaction layer increased due to the presence of the microtexture, which increased the overall surface area. This process significantly improved the interfacial diffusion reaction, thickening the interfacial IMCs layer.

The literature confirms that the presence of microtextures enhances materials’ performance in various applications. However, although there have been many studies involving the development of microtextures on material surfaces, minimal information is available on the advantages of microtextured surfaces in soldering.

A microtextured surface can affect solder alloy’s spreading area and wettability. The surface texture increases the diffusion area for the copper atom, which increases its migration to the solder alloy. Due to the diffusion and migration processes, an increased concentration of surface reaction will occur during the soldering process, which affects the microstructure and intermetallic compound (IMC) layer formation. The microstructure and IMC formation in a solder joint depends on the type of solder alloy and surface finish of the substrate [9].

The literature posits that the Sn-Cu family of alloys is preferred over other proposed Pb-free solder alloys, and Sn-0.7Cu is the best option due to its low cost and excellent performance [10,11]. The performance of the solder is important in electronic packaging, in which the solder acts as the main interconnect between a printed circuit board (PCB) and electronic components. They also provide electrical, thermal, and mechanical continuity in electronic assemblies. Simply put, the integrated circuit (IC) chip needs to be packaged in a way that it can work with bigger surroundings and function as a single system in an electronic device.

However, other researchers reported several performance problems using the Sn-0.7Cu solder alloy. For example, Gourlay et al. [12] stated that as Sn-0.7Cu solidifies, its flow ability and wettability decrease [13,14], while Mohd Salleh et al. [15] reported that Sn-0.7Cu suffers from poor microstructure refinement, which affected the solder joint’s mechanical properties.

Many researchers attempted to address these issues by incorporating alloying elements into the solder or altering its surface finish [16,17,18]. For example, Jaffery et al. [19] studied the effect of adding Iron (Fe) and Bismuth (Bi) in Sn-0.7Cu solder alloy via the resulting solder’s oxidation and wetting characteristics. They reported that the wetting properties of Sn-0.7Cu improved due to the addition of Bi and Fe into the Sn-0.7Cu solder alloy. Teoh et al. [20] reported that the mechanical strength of Sn-0.7Cu improved by adding Bi into the solder alloy. Hanim et al. [21] reported that the electroless nickel immersion silver (ENIAg) surface finish significantly enhanced the shear strength of the solder joint, as it provided a more stable metallurgical bond of the solder joint. Despite the abovementioned studies and others involving the improvement of the performance of Sn-0.7Cu, studies on the modification of the substrate used in the soldering process remain limited.

Therefore, this study looks into the fabrication of a microtexture on a substrate’s surface to improve the performance of Pb-free solder in soldering applications. Surface texturing was used to create a suitable microtexture capable of generating excellent interfacial reaction between solder alloy and copper substrate.

## 2. Materials and Methods

### 2.1. Raw Materials

A flat, square-shaped, pure copper pad with a purity of 99.9% measuring 15 mm × 15 mm × 1 mm was used as a substrate, while the Sn-0.7Cu ingot was used as a solder material.

### 2.2. Fabrication of Dimple Micro-Texture on Copper Substrate

The copper pad sample was ground using various grit sizes of silicon carbide (SiC) paper, then polished using 1 µm alumina polishing paste to remove uneven surfaces, scratches, and impurities. Next, a closed-pore dimple was developed on the polished copper surface using LST. An Ytterbium Fibre Laser Marking Machine (Herolaser; Shenzhen, China) produced a dimple micro-texture with a diameter, d, of 50 µm and a dimple spacing, S, of 100 µm. Finally, 3 samples were fabricated by varying the dimple depth, D, using different scan repetitions set at 1 to 3. The LST process parameters are summarised in Table 1, while the dimple microtexture on the copper substrate is shown in Figure 1.

### 2.3. Fabrication of Sn-0.7Cu Solder Joint

To prepare the solder ball, the Sn-0.7Cu solder ingots were melted at 350 °C in a solder pot. The ingot was cold-rolled to produce thin solder sheets, then cut into small pieces, each weighing ~0.4 g. The solder sheet was placed on a Pyrex sheet with a small amount of rosin mildly activated (RMA) flux and reflowed in a reflow oven at 250 °C and N2 gas flow. Then, the solder balls were cleaned and rinsed thoroughly using acetone and placed on the copper substrate’s surface with a small flux. Next, the reflow soldering process was used to fabricate the solder joint, using an F4N Pb-free reflow oven via a lead (Pb)-free reflow profile, per Figure 2. The solder joint was then mounted, using a mixture of epoxy resin and hardener, and cross-sectioned before the metallographic step.

### 2.4. Testing and Characterisation

#### 2.4.1. Microstructure Characterisation of the Textured Copper Substrate

The dimple-microtextured copper substrate was polished using 1 µm polishing paste for 1 min at 50 rpm to remove any resolidified material on the copper substrate. The surface morphology and roughness of the dimple-microtextured surface copper substrate were then imaged using a 3D measuring laser microscope (Model Olympus OLS5000, Shinjuku, Japan).

#### 2.4.2. Wettability of Sn-0.7Cu Solder Joint

The wettability of the solder joint was evaluated using contact angle measurement between the molten solder and copper substrate. First, the cross-sectional area of the solder joint was imaged using an optical microscope (OM), and then the contact angle was measured using Image-J.

#### 2.4.3. Microstructure Characterisation of Sn-0.7Cu Solder Joint

The solder joint’s microstructure analysis included the bulk solder’s microstructure and the thickness of the intermetallic compound (IMC) layer. The microstructural imaging of the solder substrate was carried out using OM to image the microstructure at the bulk solder. Additionally, the area along the IMC layer was measured, and its thickness was averaged per Equation (1).
IMC = Area (A)/Length (L),(1)

#### 2.4.4. Distribution of Copper Element in Solder Joint

Synchrotron Micro-X-ray Fluorescence spectroscopy and Imaging (SR-µ-XRF) was used to determine the distribution of the copper element in the solder joint. The thickness of the mounted sample was 5 mm, while its diameter was 25 mm. The samples were thoroughly cleaned using acetone in an ultrasonic cleaner to remove any impurities. The Synchrotron µ-XRF was performed at the BL6b beamline at Synchrotron Light Research Institute (SLRI), Thailand. At the beamline, a continuous synchrotron was produced from the bending magnet with an energy range of up to 12 keV and a beam size of 30 µm. The exposure time was 20 s for each spot with a step size of 0.05 mm in a helium (He) atmosphere. The data obtained were analysed using PyMca software (version 5.5.3).

#### 2.4.5. Lap Shear Strength of Sn-0.7Cu Solder Joint

A single-lap shear test was carried out using an Instron Universal Tensile Testing Machine, based on the ASTM D1002 with Cu substrate specifications of 101.6 mm × 25.4 mm × 1.5 mm. First, ~1 g of Sn-0.7Cu solder sheets were sandwiched between the two copper substrates, as illustrated in Figure 3. Next, the samples were subjected to a reflow soldering process using a tabletop Pb-free reflow oven model F4N. The samples were shear tested at a shear speed of 2 mm/min with a strain rate of 5 × 10^−4^ s^−1^, then analysed using Scanning Electron Microscope (SEM) for fracture analyses.

## 3. Results and Discussions

### 3.1. Surface Profile of the Copper Substrate

Figure 4 shows the 2D and 3D images of the dimple-microtextured copper substrates. It can be seen that a well-defined dimple shape was developed and arranged evenly on the copper surface, most likely due to the high precision control of the LST process [22]. The LST process involves heat transfer, where the material absorbs heat from laser radiation. Once the material absorbs the heat, the thermalisation process occurs, increasing the surface temperature, which promotes the melting of the copper substrates, and the melted part is subsequently ejected onto the surface. This process alters the materials’ surface morphology and topography [23].

The number of laser scan repetitions influenced the dimple depth, and it can be seen that the higher number of scan repetitions increases the dimple’s depth. For example, the average depth of a dimple for 1 scan repetition was 10 ± 5 µm, while for 2 scan repetitions, it was 30 ± 5 µm. The deepest dimple depth was for 3 scan repetitions, which was 50 ± 5 µm. During the LST process, a higher number of laser scan repetitions compelled the laser beam to stay in contact longer than a smaller number of laser scan repetitions. As the laser beam was in contact with the material longer, the material was able to absorb more laser energy, which meant more material melted and ejected to the surface, deepening the dimple depth. When a higher number of laser scan repetitions with low pulse energy were used, the ablation threshold decreased, resulting in an increase in material removal by one laser pulse [24]. As this process occurs, the increased dimple depth directly increases the surface area, thus increasing the surface roughness of the copper substrate. The surface profiles of the dimple-microtextured copper substrate with different dimple depths are shown in Figure 5.

### 3.2. Solderability of Sn-0.7Cu on Copper-Textured Surfaces

The performance of the dimple-microtextured copper substrate in soldering applications was investigated. The average contact angle of the solder joint and the copper substrates’ surface roughness is shown in Figure 6. It can be seen that the highest contact angle is for the sample with a dimple depth of 10 µm, which is 36.5°, while the lowest contact angle is for the sample with a dimple depth of 50 µm, which is 35.1°. It can also be seen that an increase in dimple depth decreased the average contact angle of the Sn-0.7Cu solder, which enhances the wettability of the Sn-0.7Cu solder. During wetting, the molten solder flowed into the valleys via capillary action [25], and Way et al. [26] stated that when the soldering process occurs, the molten solder will start melting and be attracted to the copper substrate, and capillary action pulls the solder into the dimples. However, this phenomenon disturbs the wetting process, as the molten solder must overcome the dimples’ energy barriers before thoroughly wetting the copper’s surface [27]. Therefore, as the dimple depth increases, a high volume of molten solder is needed to fill in the dimple, leading to a low volume of molten solder remaining on the surface, which decreases the contact angle.

The surface roughness of the microtextured copper substrate affects the solder joint’s average contact angle. It can be seen in Figure 6 that the highest surface roughness is for samples with a dimple depth of 50 µm, which is 8.07 µm, while the lowest surface roughness is for the samples with a dimple depth of 10 µm, which is 1.06 µm. Therefore, it can be surmised that a higher number of laser scan repetitions results in deeper dimples produced on the copper substrate. This can be attributed to the formation of dimples modifying the surface profile of the copper substrate and providing additional surface area, which increases surface roughness [28,29]. This observation is consistent with Pratap and Patra [30] and Zhang et al. [31], who reported that an increase in microtexture depth increases surface roughness.

It was observed that the surface roughness is inversely proportional to the contact angle of the solder joint, per Yulong et al. [32], who reported that the contact angle of Sn-35Bi-1Ag solder decreased when the surface roughness of the copper substrates increased, indicating that the solder wettability improved. The result is also consistent with Wu et al. [33], who reported that the substrate’s increased surface roughness significantly enhances the wetting properties.

### 3.3. Microstructure Analysis of Sn-0.7Cu Solder

The analytical results of the bulk microstructure of Sn-0.7Cu solder are shown in Figure 7. During the reflow soldering process, the copper element from the substrate tends to diffuse in the solder alloy and react with tin (Sn) due to copper diffusion. From the reactions taking place during the process, it was observed that two phases formed on the bulk area. Typically, the microstructure of Sn-0.7Cu solder alloy that forms on the substrate consists of a primary β-Sn phase or dendrites surrounded by eutectic, which is a combination of β-Sn and intermetallic Cu6Sn5 [34]. The microstructure of the solder is composed of two visible regions: eutectic and β-Sn phases. The light region represents the β-Sn phase, while the dark region represents the eutectic phase. As seen in the figure, the primary IMC increased with the dimple depth. Furthermore, it was found that the β-Sn area becomes refined and finely dispersed in the bulk microstructure. This phenomenon occurs due to the high amount of copper that dissolves in the solder alloy and reduces the β-Sn area [35]. Shen et al. [36] stated that during the solidification, the Cu atom reacts with Sn, forming intermetallic Cu6Sn5 particles and refining the grain size of the Bi-rich phase, simultaneously acting as heterogeneous nucleation sites. This finding is also similar to Hung et al. [37], who reported that the β-Sn was refined and the primary IMC increased as the copper content increased.

Figure 8 shows the micrograph of the interfacial IMC layer of the Sn-0.7Cu solder on different dimple depths of the microtextured copper substrate. It can be seen that a continuous IMC layer made up of Cu6Sn5 is present at the interface between the Sn-0.7Cu solder and copper substrates. It is also evident that the continuous layer of IMC gradually thickened as the dimple depth increased. This IMC layer will form when the solder alloy reacts with the substrate during the reflow soldering process. When the reflow soldering process begins, the element from the substrate will dissolve into the molten solder. The molten solder is then concentrated with the metals, creating a layer of IMC between the interface of the solder alloy and the substrate. The morphology of the IMC layer consists of pointed and shallow scallops, with more of the latter compared to the former in the IMC layer with increasing dimple depth. The formation of pointed scallops in the IMC layer is undesirable, as it will induce a brittle fracture at the solder joint interface compared to the shallow scallop structure [38].

The IMC thickness was averaged to determine the difference in the growth of IMC layers among dimple-microtextured copper substrates, and the results are shown in Figure 9. It can be seen that the highest average IMC is for the sample with a dimple depth of 50 µm, which is 8.8 µm, while the lowest average is for the sample with a dimple depth of 10 µm, which is 4.3 µm. It is also evident that the average thickness of the interfacial IMC layer substantially increases as the dimple depth increases. The increment in the thickness of the IMC layer for the increased dimple depth can be attributed to the influence of the surface area of the copper substrate. During wetting, chemical reactions and diffusion processes occur where the molten solder dissolves the substrate metal in a process called reactive wetting [39]. As the Sn-0.7Cu solder alloy reflows on the dimple-microtextured copper substrate, more copper diffused into the solder alloy due to the dimple microtextures on the surface providing an additional surface area that promotes diffusion and chemical reaction at the interface of the solder joint [7]. As per Dong et al. [40], due to the presence of peaks and valleys on the surface, a larger surface area was obtained between the interface of the solder joint. This stage results in the faster diffusion of the copper atoms into the solder alloy. Hence, a thick IMC layer was formed at the interface of the solder substrate. This result is supported by the distribution of copper elements in the solder joint, per Figure 10, and is discussed in the following section.

### 3.4. Distribution of Copper Element in Solder Joint

The Synchrotron Micro-X-ray Fluorescence (SR-µ-XRF) mapping was carried out to determine the distribution of copper elements in the Sn-0.7Cu solder joint. Figure 10 shows the results of SR-µ-XRF mapping. The red and blue markings on the map indicate the highest and lowest concentrations of the copper element, respectively. It can be seen that more copper elements were distributed to the solder area for the dimple-microtextured copper substrate compared to the non-textured copper substrate. The copper element, represented in red, is more evenly distributed throughout the solder balls area for the dimpled copper substrate. The highest copper concentration level in the solder ball area is shown in Figure 10b. These results prove that the dimple developed on the surface provides more surface area, increasing copper dissolution from substrates. According to Chen et al. [6], the grooves’ surface microstructure successfully enhances the diffusion mechanism where the grooves promote silver and copper atoms diffusing into titanium silico-carbide (Ti3SiC2) ceramics [41,42].

**Figure 10 materials-16-00096-f010:**
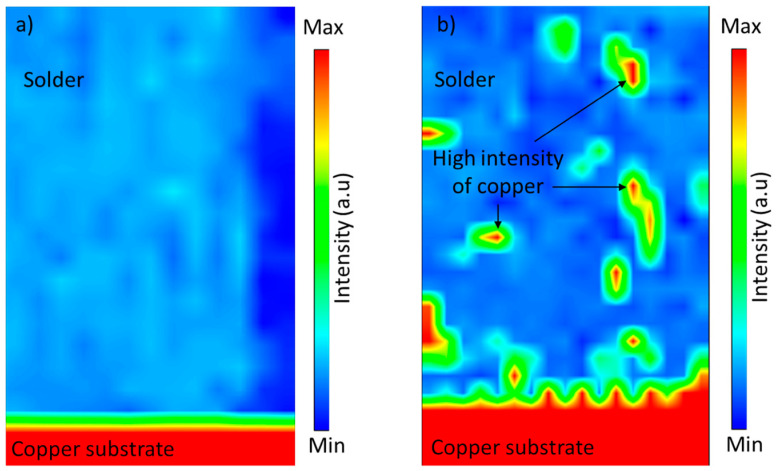
Synchrotron micro-XRF results from Cu mapping distribution of (**a**) non-textured copper substrate, (**b**) dimple-microtextured copper substrate with a dimple depth of 30 µm.

### 3.5. Lap Shear Strength of Sn-0.7Cu Solder Joint

The strength and the fracture behaviour of the Sn-0.7Cu Pb-free solder on dimple-microtextured copper substrate were determined. The shear strength of Sn-0.7Cu solder on the copper-textured surface is shown in Figure 11. It can be seen that the highest strength is for the sample with a dimple depth of 50 µm, which is 21.93 MPa, while the sample with a dimple depth of 10 µm reported the lowest strength, which is 20.93 MPa. The strength of the solder joint increases with increasing dimple depth due to the dimples on the copper substrates, which affords a greater mechanical interlocking that improves the joint bonding strength [43,44]. As a result, the higher dimple depth needed extra energy to pull the microtextures out during the debonding of the joint [45].

Figure 12 shows the fractured surface morphology obtained from SEM images for the Sn-0.7Cu Pb-free solder on a microtextured copper substrate. Based on the images, it can be surmised that the Sn-0.7Cu solder alloy has a ductile fracture mode, per the circle shape seen in the figure. The dimples are also evident in the Sn-0.7Cu solder, confirming the material’s ductile mode [46]. It can also be confirmed that there is no significant difference in the fracture behaviour of the solder joint.

## 4. Conclusions

A dimple microtexture was successfully developed on a copper substrate to enhance the surface characteristics. It was observed that increased laser scanning repetitions resulted in increased dimple depth. The deepest dimple, 50 µm, was obtained using 3 scan repetitions. In the case of the soldering application, it was confirmed that the increased dimple depth decreased the contact angle of the solder joint from 36.5° to 35.1°. It was also confirmed that the surface roughness of the copper substrates increased with increasing dimple depth. The microstructure at the bulk solder showed that more intermetallic compound Cu6Sn5 was present in the bulk solder area per increasing dimple depth. In the case of the interfacial IMC, it was revealed that the deepest dimple, which was 50 µm, has the thickest IMC layer at 8.8 µm. Through copper element mapping, the dimple microtexture on the copper substrate increased the diffusion area for the copper atom to migrate into the solder alloy. The highest strength of the Sn-0.7Cu solder alloy was reported for the sample with a dimple depth of 50 µm, which was obtained at 21.93 MPa.

## Figures and Tables

**Figure 1 materials-16-00096-f001:**
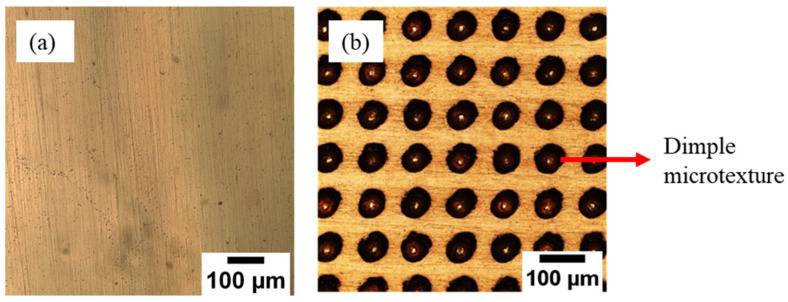
The surface of the copper substrate, (**a**) before surface texturing and (**b**) after surface texturing.

**Figure 2 materials-16-00096-f002:**
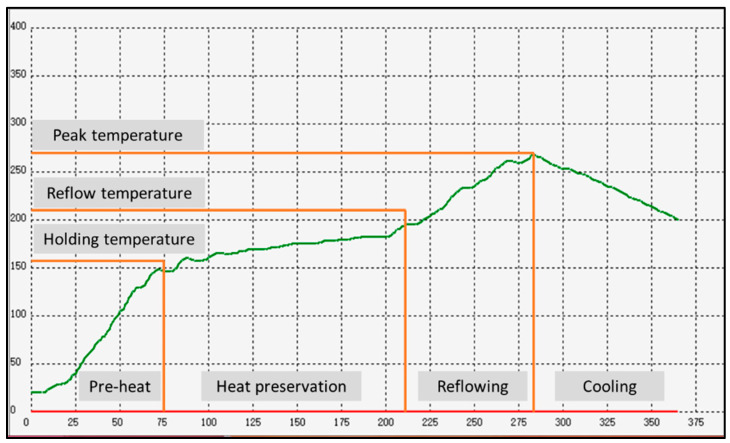
Reflow profile during the reflow soldering process.

**Figure 3 materials-16-00096-f003:**
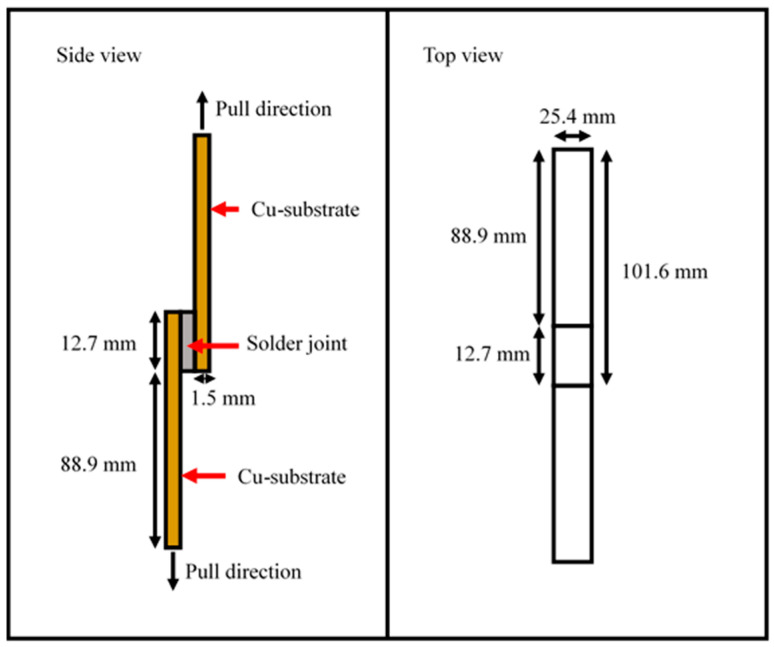
Schematic diagram of Sn-0.7Cu solder joint sample for lap-shear test.

**Figure 4 materials-16-00096-f004:**
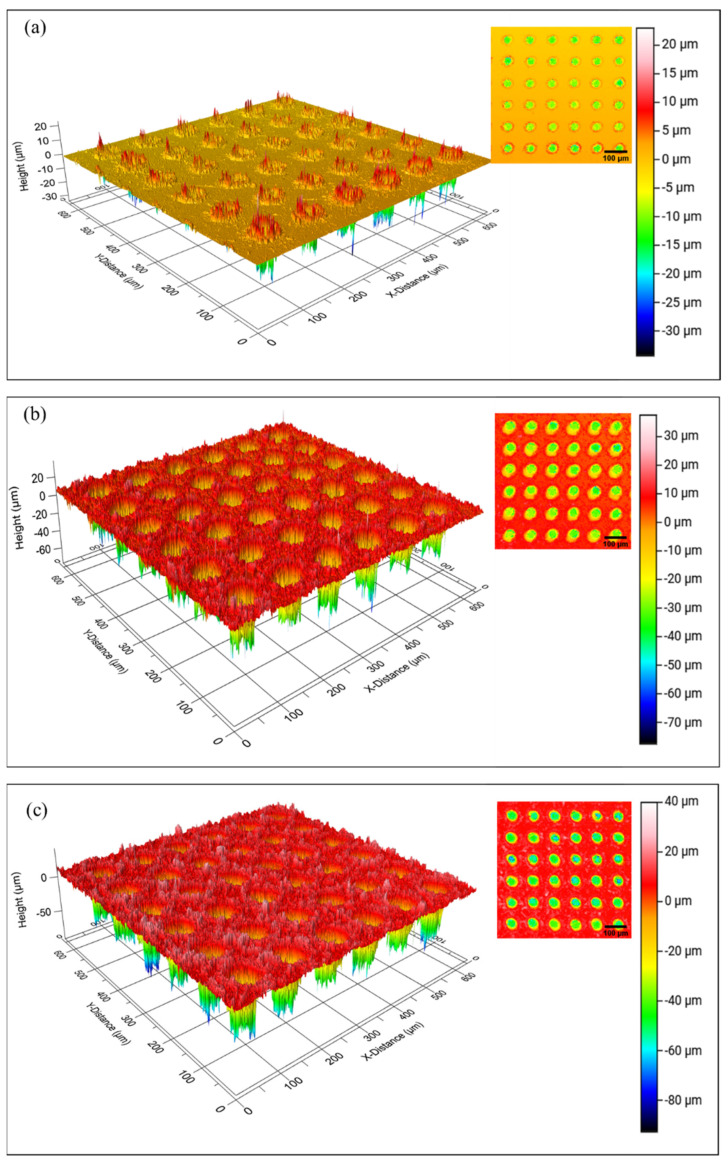
Images of the copper substrate with different depths of dimple, (**a**) 10 µm, (**b**) 30 µm, and (**c**) 50 µm.

**Figure 5 materials-16-00096-f005:**
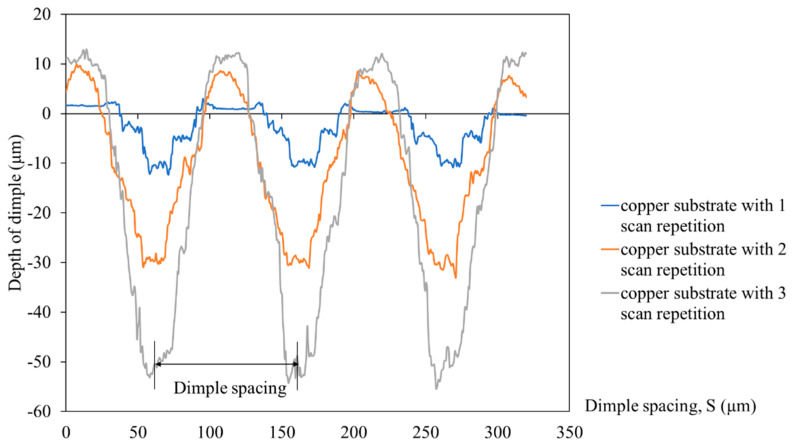
Surface profiles of the dimple−microtextured copper substrate with a different number of laser scan repetitions.

**Figure 6 materials-16-00096-f006:**
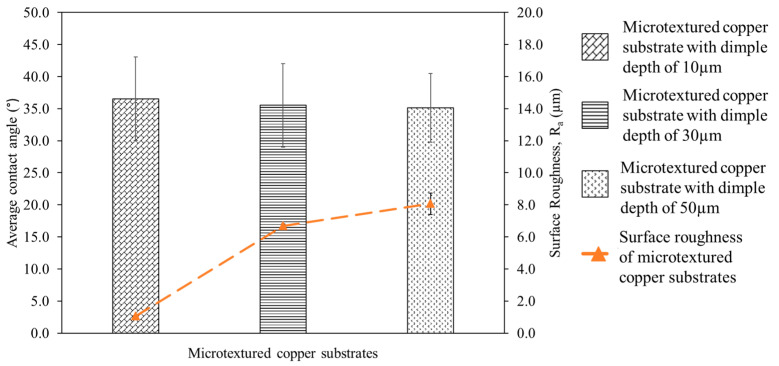
The average contact angle of the solder joint and surface roughness of microtextured copper substrates.

**Figure 7 materials-16-00096-f007:**
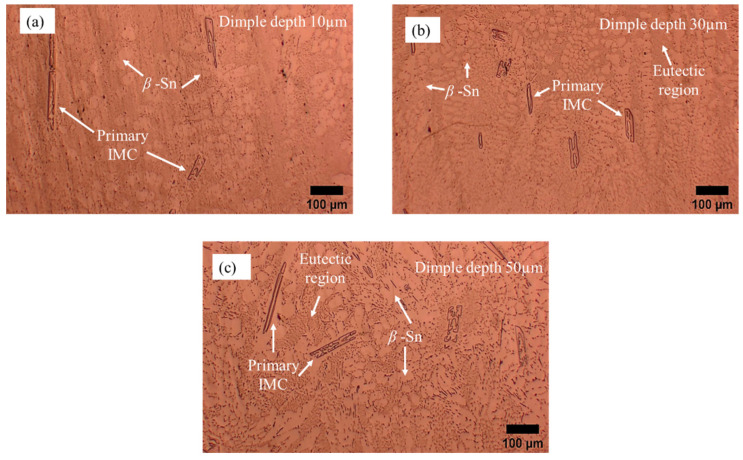
Microstructure analysis of Sn-0.7Cu solder for bulk microstructure on a microtextured copper substrate with dimple depth of (**a**) 10 µm, (**b**) 30 µm, and (**c**) 50 µm.

**Figure 8 materials-16-00096-f008:**
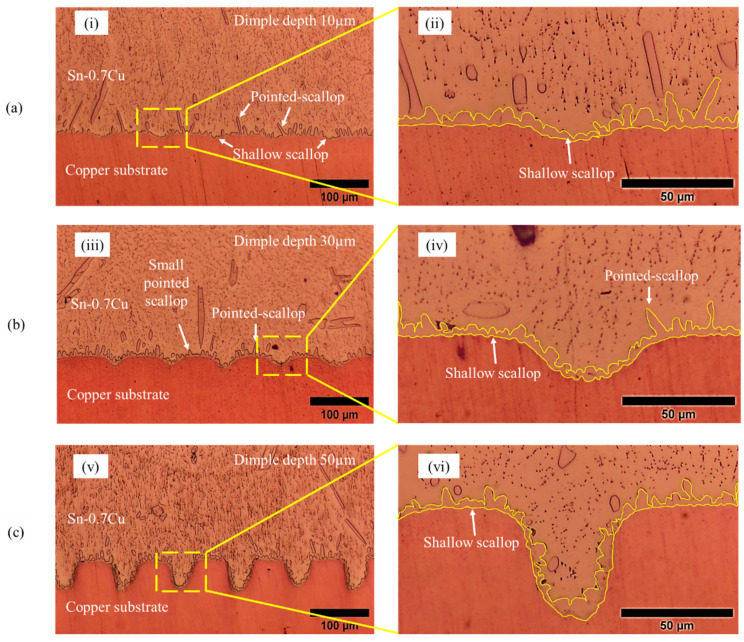
Micrograph of interfacial IMC layer on microtextured copper substrate with a dimple depth of (**a**) 10 µm, (**i**) magnification of 10×, (**ii**) magnification of 40×; (**b**) 30 µm, (**iii**) magnification of 10×, (**iv**) magnification of 40×; and (**c**) 50 µm, (**v**) magnification of 10×, (**vi**) magnification of 40×.

**Figure 9 materials-16-00096-f009:**
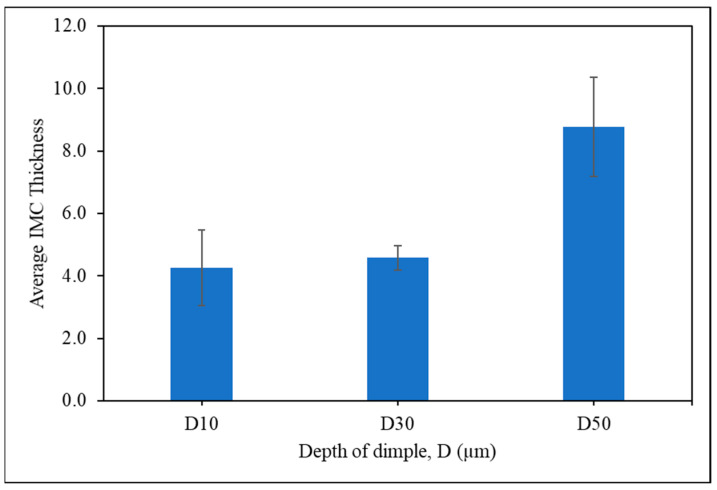
Average IMC thickness of solder joint.

**Figure 11 materials-16-00096-f011:**
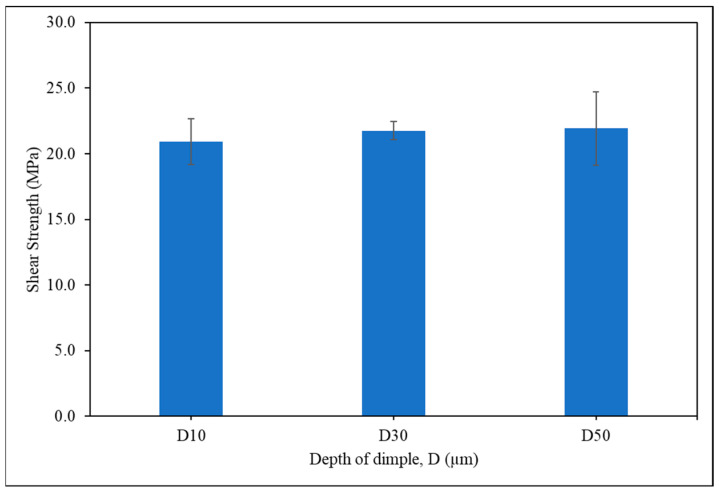
Shear strength of Sn-0.7Cu solder joint.

**Figure 12 materials-16-00096-f012:**
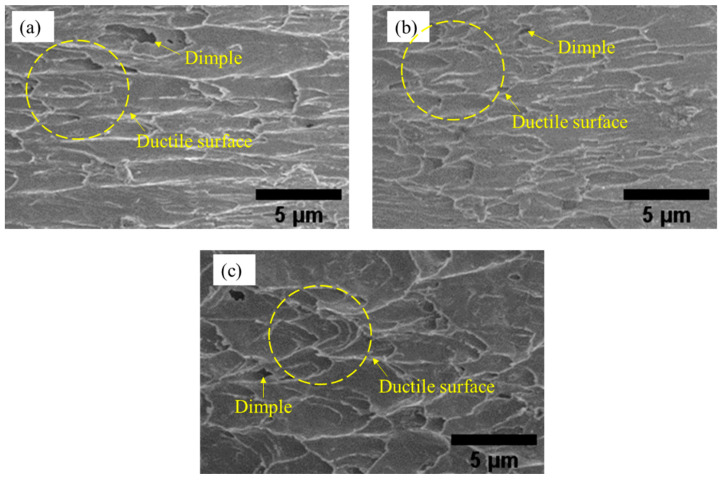
SEM images of fracture surface Sn-0.7Cu lead-free solder on different dimple depths of (**a**) 10 µm, (**b**) 30 µm, and (**c**) 50 µm.

**Table 1 materials-16-00096-t001:** Laser surface texturing parameters.

Parameter	Value
Name	Ytterbium Fiber Laser Marking Machine
Model	ML-MF-A01
Wavelength (nm)	1064
Scanning speed (mm/s)	4000
Laser power (W)	70
Frequency (kHz)	20

## Data Availability

Not applicable.

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
