# Peer review of "Effectiveness of Dimple Microtextured Copper Substrate on Performance of Sn-0.7Cu Solder Alloy"

_materials, 2022, doi:10.3390/ma16010096_

Round 1
Reviewer 1 Report
The article brings the original results regarding the investigation of dimple microtextured copper substrate on performance of Sn-0.7Cu solder alloy. Microtexture has an effect on the process of forming the interfacial interface between the substrate and the solder. I recommended the article for publication after following modifications:
- Fig. 2 a 3 – There is the illegible text in the image.
- Fig. 4 – Dependence of surface roughness - It would be appropriate to remove the connection of values using a solid line and replace it with a dashed line. Because inverse dependence probably does not exist.
- Fig. 5 – Explain the description (i), (ii), (iii) - Figure and its description should be self-supporting.
- To obtain conclusions describing the microstructure of the solder joints (Figure 5), using SEM/EDX is more suitable.
- If SEM/EDX analysis is not available, at least higher magnification would be appropriate. It is difficult to obtain values for average IMC thickness from the given results. From the image with a scale of 100 micrometers, is the result for average thickness accurate to tenths of a micrometer? (Not everything calculated can be tolerated as a valuable result).
- Expand the discussion and support it with literature – for example, in the discussion about the microstructure of solder joints, the authors support only three references (chapter "Microstructure Analysis of Sn-0.7Cu solder").
Reviewer 2 Report
In the manuscript, the investigation into the influence of the dimple microtextured copper substrate on the performance of the Sn-0.7Cu solder alloy was conducted. However, there are still some problems for the author to consider.
Comment 1: Errors in title number.
Comment 2: It is suggested that the author supplement the application background of this material to highlight the research value.
Comment 3: For introduction, the work done by scholars in this field and the relationship between this research and previous work have not been introduced.
Comment 4: In this paper, the influence of the dimple microtextured copper substrate on the performance of the Sn-0.7Cu solder alloy was discussed only from the micro level. It is suggested that the author increase the discussion on mechanical properties to further explore the influence of changes in IMC layer on mechanical properties of joints.
Comment 5: In section 2.2, please provide the reflow soldering process curve.
Round 2
Reviewer 1 Report
I recommend the article for publication.
Author Response
Thank you.
Reviewer 2 Report
In the revised manuscript, material application background and mechanical property are supplemented. However, there are still some problems to be considered by the author.
Comment 1: In introduction, there are many introductions on biomimicry products, which can be appropriately simplified.
Comment 2: It is suggested that the author introduce the application fields of the final materials. For example, aluminum alloy is used in high-speed train body to reduce weight.
